# A quasi-experimental study on stethoscopes contamination with multidrug-resistant bacteria: Its role as a vehicle of transmission

**Raeseok Lee**[1], **Su-Mi Choi**[1,2]*, **Sung Jin Jo**[3], **Songyi Han**[2], **Yun Jeong Park**[2], **Min A. Choi**[2], **Bo Kyung Kong**[4]

**1** Division of Infectious Diseases, Department of Internal Medicine, College of Medicine, The Catholic University of Korea, Seoul, Republic of Korea, **2** Department of Hospital Infection Control, Yeouido St. Mary's Hospital, College of Medicine, The Catholic University of Korea, Seoul, Republic of Korea, **3** Department of Laboratory Medicine, College of Medicine, The Catholic University of Korea, Seoul, Republic of Korea, **4** Department of Microbiology, Yeouido St. Mary's Hospital, College of Medicine, The Catholic University of Korea, Seoul, Republic of Korea

* sumichoi@catholic.ac.kr

**Data Availability Statement:** All data files are available from EPOPS database (https://doi.org/10.8888/EPOPS202101201MI).

## Abstract

Stethoscopes have been suggested to be a possible vector of contact transmission. However, only a few studies have focused on the prevalence of contamination by multidrug-resistant (MDR) bacteria and effectiveness of disinfection training to reduce. This study is to investigate the burden of stethoscope contamination with nosocomial pathogens and multidrug-resistant (MDR) bacteria and to analyze habit changes in disinfection of stethoscopes among healthcare workers (HCWs) before and after education and training. We performed a prospective pre and post quasi-experimental study. A total of 100 HCWs (55 doctors and 45 nurses) were recruited. HCWs were surveyed on their disinfection behavior and stethoscopes were cultured by pressing the diaphragm directly onto a blood agar plate before and after education on disinfection. Pulsed-field gel electrophoresis was performed to determine the relatedness of carbapenem-resistant *Enterobacteriaceae*. Most of the stethoscopes were contaminated with microorganisms before and after the intervention (97.9% and 91.5%, respectively). The contamination rate of stethoscopes with nosocomial pathogens before and after education was 20.8% and 19.2%, respectively. Stethoscope disinfection habits improved (55.1% *vs* 31.0%; *p*<0.001), and the overall bacterial loads of contamination were reduced (median colony-forming units, 15 vs 10; *p* = 0.019) after the intervention. However, the contamination rate by nosocomial pathogens and MDR bacteria did not decrease significantly. A carbapenemase-producing *Klebsiella pneumoniae* isolates from a stethoscope was closely related to isolates from the patients admitted at the same ward where the stethoscope was used. Stethoscopes were contaminated with various nosocomial pathogens including MDR bacteria and might act as a vehicle of MDR bacteria. Continuous, consistent education and training should be provided to HCWs using multifaceted approach to reduce the nosocomial transmission via stethoscopes.

**Funding:** This study was supported by grant of the Institute of Clinical Medical Research in the Yeouido St. Mary's Hospital, Catholic University of Korea (cooperative agreement no. YSI 2019-07) (to RL). The funding agencies had no role in the study design, the data collection or analysis, the decision to publish, or the preparation of the manuscript. URL: https://www.cmcsungmo.or.kr/en.common.main.main.sp.

**Competing interests:** All authors have no conflicts of interest to declare.

## Introduction

The burden of healthcare-associated infections has been steadily increasing [1]. An increasing incidence of multidrug-resistant (MDR) bacterial infection and cross-contamination with MDR bacteria is the one of the key reasons why the burden of healthcare-associated infections has increased [2]. Cross-contamination of bacteria occurs in various ways. Direct or indirect contact transmission are the most important routes of disease transmission in the hospital setting [3]. Although, hands are known to be a vector of direct contact transmission during patient care, recent studies have shown that hand hygiene alone is not enough to prevent nosocomial transmission [4]. In addition to hands, various medical devices, including blood pressure cuffs, doppler probes and even marker pens have been identified as potential vehicles of contact transmission [5–7].

The stethoscope is the most commonly used medical device that has a surface that is in direct contact with patients. Diaphragms of stethoscopes are known to be the second most contaminated area after the fingertips, even after single physical examination [8]. However, unlike hand hygiene, the role of the stethoscope as a vehicle of transmission has not yet been fully determined. Previous studies have shown that stethoscopes are contaminated mostly by gram-positive organisms such as coagulase-negative *Staphylococcus* and *Staphylococcus aureus* [9–12]. However, several studies have shown that stethoscopes may also be contaminated by gram-negative bacteria (GNB) [9, 12, 13]. A few studies have reported the proportion of resistant bacteria that were identified, and the results were limited to *S. aureus* [9–11].

The rate of stethoscope disinfection has been reported to be lower than the rate of hand hygiene [14, 15]. Only 8–35% of healthcare workers (HCWs) were disinfected their stethoscope every time, and doctors had a lower disinfection rate than nurses [11, 16]. Moreover, only 4% of HCWs performed disinfection correctly according to the standard by The US Centers for Disease Control and Prevention [14].This is partly because stethoscope disinfection is not taught or promoted to the same extent as hand hygiene. The lack of education and training is attributable to the lack of specific data on contamination by various microorganisms, and a lack of clear evidence of their role as a vehicle of transmission. The aim of this study was to investigate the burden of stethoscope contamination and the proportion of nosocomial pathogens and MDR bacteria, and to analyze habit changes in disinfection of stethoscopes, before and after education and training.

## Materials and methods

### Study design and questionnaire

We conducted a pre and post quasi-experimental study from November 6, 2018 to March 31, 2019 at a 450-bed university-affiliated teaching hospital in Seoul, Korea (Fig 1). Among doctors and nurses working in the general wards, intensive care units and emergency room, only those who voluntarily agreed to participate in this study were enrolled. The medical department and ward included the departments of internal medicine and pediatrics. The surgical

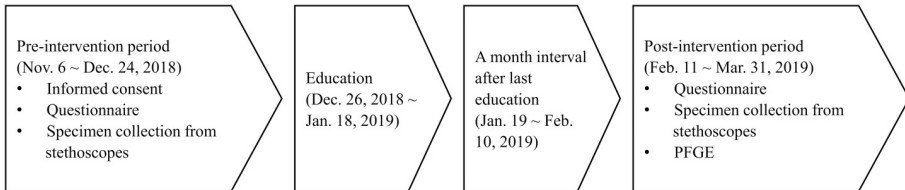

**Fig 1. Study flow for pre and post quasi-experimental study.** PFGE, Pulsed-field gel electrophoresis.

department and ward included the departments of general surgery, neurosurgery and obstetrics and gynecology. After obtaining informed consent, a pre-intervention questionnaire was administered, and specimens were collected from stethoscopes. The staff of the department of hospital infection control conducted a training and education on stethoscope disinfection methods and practices, separately by divisions of participants, using flyers and PowerPoint slides (Fig 1). The training and education were conducted twice for each participant, one hour at a time. The post-intervention questionnaire administration and collection of specimens from stethoscopes was carried out by the same investigator, one month after the last education session. All samples for culture were collected without prior notice. Various parameters potentially associated with stethoscope contamination were included in the survey. The questionnaire included (1) demographic data such as sex, age group, specialty, work department, and length of career and; (2) stethoscope disinfection habits such as frequency, preferred methods, preferred places, and frequency of hand washing. The questionnaire was mainly created based on previous studies, but several items were added and modified after study meeting [11, 17, 18]. Furthermore, the questionnaire was pre-tested by three doctors and two nurses who did not participate in the study and finally revised before the study (S1 and S3 Files). This study was approved by the institutional review board of the Yeouido St. Mary's Hospital (No. SC18OESI0120 and No. SC19RESI0081). All participants provided written informed consent.

## Specimen collection and culture methods

Personal stethoscopes of doctors and shared stethoscopes used by nurses in the wards were included. Culture specimens from stethoscopes were obtained by pressing the diaphragm directly onto a blood agar plate for 6 to 10 seconds by a trained investigator. The inoculated plates were incubated aerobically at 37°C for 24 hours. Colony-forming units (CFUs) were identified and counted by a single microbiologist. Bacterial identification and antimicrobial susceptibility tests were performed using the Vitek 2 automated system (bioMérieux, Marcy L'Étoile, France) following the manufacturer's recommendations. The results of the susceptibility test were interpreted based on the guideline document M100-S28 by Clinical and Laboratory Standards Institute [19]. The modified carbapenem inactivation method was used as a phenotypic test for carbapenem-resistant *Enterobacteriaceae*. The BD MAX system (BD Molecular Diagnostics, Franklin Lakes, NJ, USA) with Check-Direct carbapenemase-producing *Enterobacteriaceae* (CPE) real-time PCR was used for the detection of carbapenemase genes according to the manufacturer's instructions.

We defined potential nosocomial pathogens as follows: *S. aureus*, Enterococcus species, *Pseudomonas aeruginosa*, Acinetobacter species, and *Enterobacteriaceae*. MDR was defined as non-susceptibility to one or more agents in at least three different antibiotic classes among gram-negative organisms and methicillin-resistant staphylococci and vancomycin-resistant enterococci [20].

## Pulsed-field gel electrophoresis

During the study period, we were collecting CPE clinical isolates from patients for other purposes. We performed pulsed-field gel electrophoresis (PFGE) to determine the genetic relatedness of CPE isolated from the stethoscope and patients admitted to the ward where the stethoscope was used during study period. Genomic DNA extracted from the isolates were digested by *XbaI* restriction endonuclease (Roche, Mannheim, Germany) and DNA fragments were separated with a running time of 20 hr at 6 V/cm at 11°C on Genemapper X (Bio-Rad, Hercules, CA) with initial and final pulse times of 0.5 sec and 30 sec, respectively. A lambda ladder (Bio-Rad) was used as a DNA size marker. Similarity coefficients were calculated from

Dice coefficients. Banding patterns were analyzed with BioNumerics software version 6.0 (Applied Maths, Sint-Martens-Latem, Belgium) to make a dendrogram. Organisms with PFGE profiles with greater than 90% similarity were considered genetically related strains.

## Statistical analysis

Categorical variables were described as frequencies and proportions, and continuous variables were reported using the mean and standard deviation, or the median and interquartile range. Categorical variables were compared by using the $\chi^2$-test or Fisher's exact test. The non-parametric McNemar's test for categorical variables and Wilcoxon signed-rank test for continuous variables were used for pre- and post-intervention comparisons. Multivariate logistic regression was performed to identify risk factors for contamination by nosocomial pathogens using all variables with a P-value of less than 0.1 in the univariate analysis and clinically important variables. Before the multivariate regression, multicollinearity was assessed using variance inflation factors. All statistical analyses were performed using SAS version 9.4 (SAS Institute, Cary, NC) with a two-sided P-value of less than 0.05 considered significant.

## Results

### Demographic characteristics and pre-intervention status of the participants

One hundred participants, comprising 55 doctors and 45 nurses were enrolled, of whom 59 were women. However, the proportion of women was lower among doctors (29.1%, 16/55) and higher among nurses (95.6%, 43/45). Most of the participants (88.0%) were aged less than 40 years and 84.0% had used stethoscopes for more than one year. In the pre-intervention assessment, 85.0% reported that they had been educated and had an accurate knowledge of hand hygiene, but only 22.0% reported that they had been educated about the need for stethoscope disinfection. The demographic characteristics and pre-intervention education status of the participants are shown in Table 1.

### Stethoscope disinfection habits before and after intervention

Before the intervention, 88.0% of the participants responded that they believed that disinfection of stethoscopes would reduce contact transmission. However, only 31.0% reported that they regularly–defined as for every patient to at least once a week–disinfected their stethoscope. This rate was higher in nurses (44.4%, 20/45) than doctors (20.0%, 11/55) (odds ratio [OR], 0.313; 95% confidence interval [CI], 0.129–0.757; $p = 0.008$), Also, the rate of disinfection of stethoscopes at least once a week was higher in women (39.0%, 23/59) than men (19.5%, 8/41) (OR, 0.379; 95% CI, 0.149–0.965; $p = 0.038$). There were no significant differences according to age or length of career. Overall, the use of alcohol swabs was the preferred methods of disinfection. Twelve of the doctors, but none of the nurses, used the alcohol-based hand gel provided next to the patients' beds (Table 2).

After the education on disinfection, the proportion of participants who reported that they disinfected their stethoscope at least once a week significantly increased (31.0% [31/100] vs 55.1% [54/98]; $p<0.001$). This was consistent in men (19.5% [8/41] vs 37.5% [15/40]; $p = 0.039$) and women participants (39.0% [23/59] vs 67.2% [39/58]; $p<0.001$), and in doctors (20.0% [11/55] vs 41.5% [22/53]; $p = 0.001$) and nurses (44.4% [20/45] vs 71.1% [32/45]; $p = 0.004$). The participants' preferred methods of disinfection did not change after the education and training, and alcohol swabs were the most frequently used method. Most of the participants reported that the education and training had been helpful and effective in changing

**Table 1. Demographic characteristics and pre-intervention status of the participants.**

| Characteristics | Total N = 100 (%) |
|---|---|
| Sex, woman | 59 (59.0) |
| Age, years | |
| 20–29 | 31 (31.0) |
| 30–39 | 57 (57.0) |
| 40–49 | 9 (9.0) |
| ≥50 | 3 (3.0) |
| Subject | |
| Doctor | 55 (55.0) |
| Medical department | 38 (69.1) |
| Surgical department | 7 (12.7) |
| Emergency medicine | 4 (7.3) |
| Intern | 6 (10.9) |
| Nurse | 45 (45.0) |
| Medical ward | 20 (44.5) |
| Surgical ward | 6 (13.3) |
| Emergency room | 4 (8.9) |
| Intensive care unit | 15 (33.3) |
| Career period, years | |
| <2 | 29 (29.0) |
| 2–5 | 32 (32.0) |
| 5–10 | 18 (18.0) |
| ≥10 | 21 (21.0) |
| Period of stethoscope use | |
| <6 months | 7 (7.0) |
| 6 months to 1 year | 9 (9.0) |
| ≥1 year | 84 (84.0) |
| Educational experiences in stethoscope disinfection | |
| Need for disinfection | 22 (22.0) |
| during college education | 2 (2.0) |
| by hospital infection control unit | 15 (15.0) |
| by senior or colleague | 5 (5.0) |
| Methods of disinfection | 16 (16.0) |
| during college education | 1 (1.0) |
| by hospital infection control unit | 12 (12.0) |
| by senior or colleague | 3 (3.0) |
| Use a personal stethoscope to auscultate patients with MDR pathogens | 36 (36.0) |
| Reasons of not using designated stethoscope | |
| Ignorance of designated stethoscope | 3 (3.0) |
| Hard to access | 5 (5.0) |
| Function is not good | 19 (19.0) |
| Exact knowledge of hand washing | 85 (85.0) |
| Time of hand washing | |
| <20 sec | 61 (61.0) |
| 20–40 sec | 34 (34.0) |
| ≥40 sec | 5 (5.0) |

MDR, Multidrug-resistant

**Table 2. Disinfection habit changes of the participants after education.**

| Question | Habit and practices | Pre-intervention | Post-intervention |
|---|---|---|---|
|  |  | N = 100 (%) | N = 98 (%)[1] |
| Helpful to change disinfection habit | Yes |  | 64 (65.3) |
| Frequency of disinfection |  |  |  |
|  | Every patient | 10 (10.0) | 20 (20.4) |
|  | At least once a week | 21 (21.0) | 34 (34.7) |
|  | On occasion | 51 (51.0) | 40 (40.8) |
|  | Not at all | 18 (18.0) | 4 (4.1) |
| Method of disinfection |  |  |  |
|  | Alcohol swab | 75 (75.0) | 78 (79.6) |
|  | Alcohol-based hand gel | 12 (12.0) | 19 (19.4) |
|  | Soap and water | 0 (0.0) | 0 (0.0) |
| Place of disinfection |  |  |  |
|  | Patient's bed side | 36 (36.0) | 45 (45.9) |
|  | Nurse office | 42 (42.0) | 40 (40.8) |
|  | Outpatient room | 9 (9.0) | 11 (11.2) |

[1]Two participants moved to another hospital after the pre-intervention samples.

their disinfection habits. Their self-reported changes in disinfection practices are summarized in Table 2.

## Burden of contamination of stethoscopes and nosocomial pathogens identified

Four participants were excluded from the contamination analysis because they changed their stethoscope during the study period and an additional two participants were excluded from the post-intervention analysis because they moved to other hospital during the study period.

Before the intervention, 94 of 96 stethoscopes (97.9%) were contaminated with at least one microorganism, and there was a median of 15 CFUs (interquartile range: 5–36) per stethoscope. The bacterial load was significantly lower in the stethoscopes of participants who disinfected their stethoscope at least once a week than that of participants who disinfected their stethoscope less frequently or did not disinfect them at all (median CFUs, 9 *vs* 19; *p* = 0.015), nurses than doctors (median CFUs, 10 *vs* 20; *p* = 0.018) and women than men (median CFUs, 13 *vs* 29; *p* = 0.017). However, the bacterial load did not differ significantly according to the participant's career duration, method of stethoscope disinfection, or duration of stethoscope use. Twenty-two potential nosocomial pathogens were isolated from 20 of 96 stethoscopes (20.8%) (Table 3). Three (13.6%) were MDR organisms consisting of two methicillin-resistant *S. aureus* and one extended-spectrum beta-lactamase producing *Klebsiella pneumoniae* isolate.

After the intervention, 86 of 94 (91.5%) stethoscopes were contaminated, and the load of contaminated bacteria was a median of 10 CFUs (interquartile range: 3–35). The overall bacterial load was significantly lower in the post-intervention period than in the pre-intervention period (median CFUs 15 vs 10; *p* = 0.019). However, the reduction was only observed in the stethoscopes of women and nurses, and not in the stethoscopes of men and doctors (Fig 2). The overall contamination rate was reduced but it was not statistically significant (97.9% vs 91.5%; *p* = 0.073) and the contamination rate by potential nosocomial pathogens and MDR bacteria did not decrease. Similar to before the intervention, 22 potential nosocomial pathogens were isolated from 18 (19.2%) of 94 stethoscopes of which 7 (31.8%) were MDR

**Table 3. Isolated nosocomial pathogens and proportion of MDR organisms.**

|  | Pre-intervention N = 96 (%) | | Post-intervention N = 94 (%) | |
| --- | --- | --- | --- | --- |
|  | Pathogens | MDR | Pathogens | MDR |
| No. of stethoscope | 20 (20.8) | 3 (3.1) | 18 (19.2) | 6 (6.4) |
| *Staphylococcus aureus* | 13 (13.5) | 2 (2.1)[1] | 15 (15.7) | 4 (4.3) |
| Enterococcus species | 6 (6.3) | 0 (0.0) | 4 (4.3) | 0 (0.0) |
| *Acinetobacter baumannii* | 0 (0.0) | 0 (0.0) | 1 (1.1) | 1 (1.1) |
| *Pseudomonas aeruginosa* | 0 (0.0) | 0 (0.0) | 0 (0.0) | 0 (0.0) |
| *Enterobacteriaceae* | 3 (3.1) | 1 (1.2) | 2 (2.1) | 2 (2.1) |
| *Klebsiella pneumoniae* | 1 (1.2) | 1 (1.2)[2] | 1 (1.1) | 1 (1.1)[3] |
| *Escherichia coli* | 0 (0.0) | 0 (0.0) | 1 (1.1) | 1 (1.1)[2] |
| *Enterobacter cloacae* | 2 (2.3) | 0 (0.0) | 0 (0.0) | 0 (0.0) |

MDR, Multidrug-resistant

[1]Methicillin-resistant *Staphylococcus aureus*

[2]Extended-spectrum beta-lactamase producer

[3]Carbapenemase-producing *Enterobacteriaceae*

organisms (Table 3). One isolate of *K. pneumoniae* was found to be carbapenem-resistant. Other than bacteria, *Aspergillus fumigatus*, *A. niger* and a mold were cultured in the pre-intervention cultures and three molds were cultured in the post-intervention cultures. These fungi were not included in the statistical analysis and reporting of the colony counts.

The risk of contamination with nosocomial pathogens was independently associated with a high bacterial load as the result of the multivariate analysis with sex, department of participants, frequency and methods of disinfection, and period of stethoscope use during both the pre- and post-intervention period (adjusted odds ratio, 1.016; 95% CI, 1.000–1.031; $p$ = 0.049 and adjusted odds ratio, 1.033; 95% CI, 1.006–1.060; $p$ = 0.015, respectively).

## PFGE for CPE

During post-intervention period, a carbapenemase-producing *K. pneumoniae* was isolated from a stethoscope shared in the ward. During the period preceding the isolation of the carbapenemase-producing *K. pneumoniae* from the stethoscope, three isolates of carbapenemase-producing *K. pneumoniae* from three patients admitted to the ward in which the stethoscope

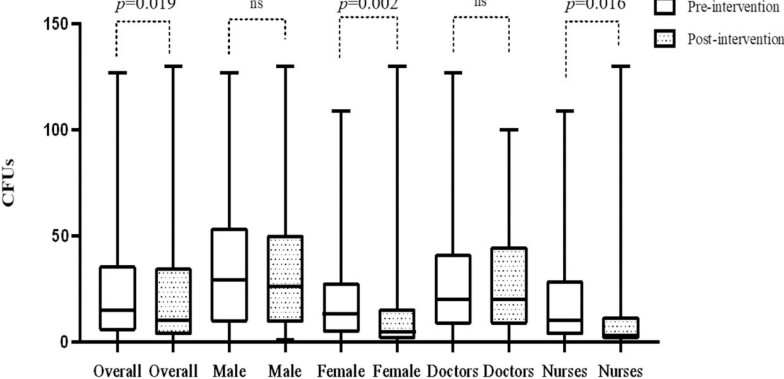

**Fig 2. Changes in colony forming units of bacteria isolated from stethoscopes during pre- and post-intervention period.** CFUs, colony forming units; ns, non-specific.

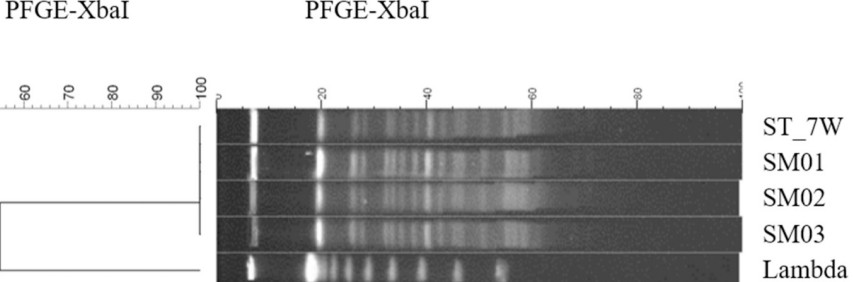

**Fig 3. Result of pulsed field gel electrophoresis and dendrogram of carbapenemase-producing *K. pneumoniae*.**
Percentage similarities are shown above the dendrogram. ST_7W, *K. pneumoniae* from the stethoscope; SM 01 to 03, *K. pneumoniae* isolates form the patients. PFGE, Pulsed-field gel electrophoresis.

was used, were available for PFGE. Two isolates were cultured from rectal swabs 6 weeks prior to the stethoscope culture and another was cultured in a blood sample 5 weeks prior to the stethoscope culture. All isolates showed the same antibiotic susceptibility and had KPC-2 in real-time PCR. *K. pneumoniae* isolates from the stethoscope and the patients belonged to one cluster pattern with a similarity of over 95% on PFGE (Fig 3 and S1 Fig).

## Discussion

In this study, we found most of the stethoscopes were contaminated with microorganisms. Twenty percent of the stethoscopes were contaminated with potential nosocomial pathogens, and 3–6% with MDR bacteria. After the intervention, the participants reported changes in their stethoscope disinfection habits, and overall bacterial loads of contamination were lowered. However, the rate of contamination with nosocomial pathogens and MDR bacteria did not decrease. A CPE from stethoscope showed a genetic relatedness with the isolates from patients admitted at the ward in which the stethoscope was used.

Noncritical items in the medical environment such as bed rails, bed side tables, blood pressure cuffs, monitors can be the source of healthcare-associated infections [21]. There is evidence that thorough medical environmental disinfection reduces healthcare-associated infections [22]. However, there is disagreement regarding the role of stethoscopes as a vehicle of transmission of nosocomial pathogens and MDR bacteria [9, 13, 23, 24]. According to previous studies, gram-positive bacteria were the most prevalent contaminants of stethoscopes and other medical equipment [9, 10, 25]. GNB were known to rarely contaminate stethoscopes [9, 12, 18]. Although, we did not identify all isolated microorganisms as species, GNB were not uncommon. Six of all 44 nosocomial pathogens cultured from stethoscopes were GNB, which was similar to those published by Knecht *et al* [13] using RNA sequencing. Methicillin-resistant *S. aureus* was the only microorganism reported to be resistant bacteria that contaminate stethoscopes and methicillin-resistant rate varied greatly from 0 to 42% [9, 11]. To our knowledge, the rate of MDR GNB contamination has not been reported previously. In this study, four of the six (66.7%) potential nosocomial GNB were MDR bacteria. Notably, the MDR rate was even higher in GNB than in gram-positive bacteria in both the pre- and post-intervention period. However, the absolute total number of GNB is so small, further study in the larger groups should be needed.

The US Centers for Disease Control and Prevention recommends disinfecting noncritical patient-care devices such as stethoscopes and blood pressure cuffs when they are visibly soiled and on a regular basis such as after use on each patient, once daily, or once weekly [26]. However, in actual clinical practice, less than a quarter of the participants reported that they had

been educated to disinfect stethoscopes, and less than a third reported that they had disinfected their stethoscope at least once a week. Similar results have been found in previous studies using self-reported questionnaires [16, 18] and these rates might be overreported because the frequency has been only 15 to 18% in studies that used direct observation [14, 15]. To increase the frequency of stethoscope disinfection, disinfection education was done using flyers and PowerPoint presentations for doctors and nurses for 3 weeks. Post-intervention stethoscope culture was performed a month after the final education session to assess whether the participants maintained the disinfection practices that had been taught. There are opposing views on the effectiveness of education or training [23, 24]. In our study, the self-reported frequency of disinfection increased and the total CFUs of the contaminants were significantly reduced. Although not statistically significant, the proportion of stethoscopes contaminated with microorganisms decreased.

There might be several reasons for this result. First, we spent equal time and effort on education and training all participants, rather than focusing on a particular group known to have low disinfection rates. Men and doctors have previously been shown to have relatively low disinfection rates and the same results were observed in our study [10, 16]. Moreover, they showed a relatively limited improvement, even after education. Second, education session and a short training period were insufficient to change the behavior of HCWs. Third, instead of a team-based approach, only hospital infection control staff participated in education and training as educators. The increase in the self-reported disinfection rates and reduction of total bacterial loads was insufficient to reduce the opportunity of stethoscope as a transmission vehicle of pathogens. Continuous efforts are needed in the multifaceted aspect to improve HCWs' behavior along with repeated education and training [27, 28]. Lastly, the limitation of alcohol-based disinfection methods used by most HCWs. Alcohol-based disinfection methods are widely recommended as its effectiveness and convenience [10]. However, it evaporates quickly, and residual effect can not be expected. Effect of disinfection would be reduced if disinfection is not performed after each contact with the patient.

Although there was a gap of 5 to 6 weeks, one isolate of carbapenemase-producing *K. pneumoniae* from a stethoscope and three clinical isolates from patients showed one cluster pattern. This suggested that stethoscope might be contaminated during patient care and demonstrated that nosocomial pathogens can survive for more than a few weeks on the inanimate surface, as is well known [29]. Because stethoscope might be a vehicle of nosocomial pathogens even MDR bacteria, we emphasize that disinfecting stethoscope is as important as hand hygiene.

This is the first study to evaluate the contamination rate of stethoscopes by GNB-MDR bacteria and to demonstrate the possibility of stethoscope as a vehicle for MDR bacteria such as CPE, but with several limitations. Since this study was conducted in a single hospital, the generalization of the results is limited. Due to the small number of study participants, a more detailed subgroup analysis could not be performed. Second, we chose the direct imprinting method as the contamination level of the diaphragm is higher than that of other parts of the stethoscope after performing physical examinations [8, 30], and direct imprinting method was known to be more efficient than collecting samples with a sterile cotton swab [9]. When we checked the blood agar plates on which the diaphragm was pressed, most of the bacterial colonies were detected only in urethane rim area. For this reason, the level of contamination might have been underestimated. However, as the peripheral rim was found to be the most heavily contaminated area of the stethoscope, the result can be expected to be close to the actual level of contamination [31]. Though our study only provided the effectiveness of education and the traditional disinfection methods, further study about the usefulness with a variety of options such as barriers, ultraviolet disinfection, and disposable stethoscope is needed [17, 32].

In conclusion, stethoscopes were contaminated with various nosocomial pathogens including MDR bacteria and could cause cross-transmission in hospital setting. Accurate and practical guidelines for disinfection of stethoscopes based on evidence are needed. Also, education and training to the HCWs must be conducted on an ongoing basis, using an intensive multidisciplinary and team-based approach.

## Supporting information

**S1 Fig. Raw image of pulsed field gel electrophoresis of carbapenemase-producing** *K. pneumoniae.* M, lambda ladder; ST_7W, *K. pneumoniae* from the stethoscope; SM 01 to 03, *K. pneumoniae* isolates form the patients.
(PDF)

**S1 File.**
(PDF)

**S2 File.**
(PDF)

**S3 File.**
(PDF)

**S4 File.**
(PDF)

## Acknowledgments

We would like to express our gratitude to Dr. Jong-Sung Lim at National Instrumentation Center for Environmental Management for his technical assistance for PFGE.

## Author Contributions

**Conceptualization:** Raeseok Lee, Su-Mi Choi.

**Data curation:** Raeseok Lee, Songyi Han, Yun Jeong Park, Min A. Choi.

**Formal analysis:** Sung Jin Jo, Bo Kyung Kong.

**Funding acquisition:** Raeseok Lee.

**Investigation:** Raeseok Lee, Sung Jin Jo, Songyi Han, Yun Jeong Park, Min A. Choi, Bo Kyung Kong.

**Methodology:** Su-Mi Choi.

**Project administration:** Su-Mi Choi.

**Resources:** Songyi Han, Yun Jeong Park, Min A. Choi.

**Supervision:** Su-Mi Choi.

**Writing – original draft:** Raeseok Lee.

**Writing – review & editing:** Su-Mi Choi, Sung Jin Jo.

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
