## [Decision Letter · Decision Letter 0]

16 Dec 2020

PONE-D-20-26077

A quasi-experimental study on stethoscopes contamination with multidrug-resistant bacteria: Its role as a vehicle of transmission

PLOS ONE

Dear Dr. Choi,

Thank you for submitting your manuscript to PLOS ONE. After careful consideration, we feel that it has merit but does not fully meet PLOS ONE’s publication criteria as it currently stands. Therefore, we invite you to submit a revised version of the manuscript that addresses the points raised during the review process.

I would like to sincerely apologise for the delay you have incurred with your submission. It has been exceptionally difficult to secure reviewers to evaluate your study. We have now received three completed reviews; their comments are available below. Reviewer#1 and #3 have raised some concerns about the study that need to be addressed in a revision. Please provide more details on how the sample size was calculated, the sampling procedure and the use of training as an intervention.

We look forward to receiving your revised manuscript.

Kind regards,

Miquel Vall-llosera Camps

Senior Editor

PLOS ONE

Journal Requirements:

2. Please provide additional details regarding participant consent.

In the ethics statement in the Methods and online submission information, please state what type of consent you obtained (for instance, written or verbal, and if verbal, how it was documented and witnessed).

If your study included minors, state whether you obtained consent from parents or guardians.

If the need for consent was waived by the ethics committee, please include this information.

3. Please include additional information regarding the pre- and post-intervention questionnaire used in the study and ensure that you have provided sufficient details that others could replicate the analyses.

For instance, if you developed a questionnaire as part of this study and it is not under a copyright more restrictive than CC-BY, please include a copy, in both the original language and English, as Supporting Information.

**Comments to the Author**

1. Is the manuscript technically sound, and do the data support the conclusions?

Reviewer #1: Yes

Reviewer #2: Yes

Reviewer #3: Partly

2. Has the statistical analysis been performed appropriately and rigorously? 

Reviewer #1: Yes

Reviewer #2: Yes

Reviewer #3: I Don't Know

3. Have the authors made all data underlying the findings in their manuscript fully available?

Reviewer #1: Yes

Reviewer #2: Yes

Reviewer #3: Yes

4. Is the manuscript presented in an intelligible fashion and written in standard English?

Reviewer #1: Yes

Reviewer #2: Yes

Reviewer #3: Yes

5. Review Comments to the Author

Reviewer #1: This is a well-conducted study regarding the effect of an educational program as an intervention to decrease the contamination rate of stethoscopes in use within a hospital. However, in this regard I have some observations:

1. Although the program was shown to have a positive effect on cleaning habits and cleaning frequency in stethoscopes, no effect on the contamination rate was observed. The authors focus the discussion of this point on aspects related to the training program; however, they do not consider the nature of the disinfectant used, which could also have an effect on the contamination rate. The disinfectant reported by the participants was ethyl alcohol, which has a good effect on bacteria, fungi and even viruses; however, it evaporates quickly, allowing re-contamination of the stethoscope in the time after cleaning, where the stethoscope can be in contact with the patient's environment or the patient himself.

2. The authors make the following statement in the discussion: “This is the first study to evaluate the contamination rate of stethoscopes by nosocomial pathogens and MDR bacteria and to demonstrate the possibility of stethoscope as a vehicle for MDR bacteria such as CPE, but with several limitations. " This statement is not entirely correct and in any case, they must show evidence to support such statement.

3. In table 3 the names of the isolated bacteria must be written according to the binomial system, following the nomenclature rules for publication in scientific journals.

Reviewer #2: This is an extremely well written article in a challenging area of research that has never been published before. They evaluated the potential for the stethoscope to trasmit multidrug resistant (MDR) organisms, as well as the impact of education on stethoscope hygiene. While the latter has previously been shown to have little impact (and is consistent with the authors findings), the former is uniquely important, as it clearly demonstrate the potential for the stethoscope to serve as a MDR organism vector. This paper will be frequently cited, and will serve as an important contribution to the literature. I have reviewed over 1000 articles for various journals (including PLOS ONE) in my career. I have NEVER recommended acceptance without a number of suggestions. The only suggestion I have for this manuscript is that they suggest methods that may decrease the potential for MDR transmission by the stethoscope. There are a number of stethoscope hygiene strategies on the market that include barriers, caps, UV light, and disposable stethoscopes (although these are terrible tools) that do not require health care practioner cleaning (which has been repetitively proven to be an ineffective intervention). It would be reasonable to suggest them here. Further, it is a covid world, and the potential for the stethoscope being a vector may have greater impact than when this reasearch started. Here are a few citations. Thank you for the privilege of reviewing this excellent manuscript. New Scope for the Stethoscope. Kalra S, Reddy S. Mayo Clin Proc Innov Qual Outcomes. 2020 Feb 5;4(1):1-2. Aseptic Barriers Allow a Clean Contact for Contaminated Stethoscope Diaphragms. Vasudevan R, Shin JH, Chopyk J, Peacock WF, Torriani FJ, Maisel AS, Pride DT. Mayo Clin Proc Innov Qual Outcomes. 2020 Feb 5;4(1):21-30. The stethoscope: a potential vector for COVID-19? Vasudevan RS, Bin Thani K, Aljawder D, Maisel S, Maisel AS. Eur Heart J. 2020 Sep 21;41(36):3393-3395. Comparing the auscultatory accuracy of health care professionals using three different brands of stethoscopes on a simulator. Mehmood M, Abu Grara HL, Stewart JS, Khasawneh FA. Med Devices (Auckl). 2014 Aug 14;7:273-81.

Reviewer #3: I just congratulate all the authors for this interesting study.

Line 22: please include a paragraph that describe HCWs stethoscope disinfection practice of HCWs from previously conducted studies and the challenges behind/ or the driving force/. Use the following article as an example and many others from PubMed

Ex:" Sahiledengle B. Stethoscope disinfection is rarely done in Ethiopia: What are the associatedfactors?PLoSONE.2019: 14(6):e0208365. https://doi.org/10.1371/journal.pone.0208365 "

Line 36: Does the authors included all HCWs from the specific dep't ?

Line 39: self-administered or what ? what about the validity and reliability of the data collection tool ?

line 45: "conducting a training for a month" is not clear: does the training given for 30 days? or how many days; how many hours ? when ? should be clear

Line 95: We did not see a multivariate logistic regression table in the manuscript ?

6. PLOS authors have the option to publish the peer review history of their article (what does this mean?). If published, this will include your full peer review and any attached files.

Reviewer #1: **Yes: **José A. Alvarez-Canales

Reviewer #2: **Yes: **W. Frank Peacock, MD, FACEP, FACC, FESC

Reviewer #3: **Yes: **Biniyam Sahiledengle

---

## [Author Response · Author response to Decision Letter 0]

13 Mar 2021

We would like to thank editor and all reviewers for the invaluable comments and advices that help a great deal in improving our original manuscript.

---

## [Editor Report · Decision Letter 1]

7 Apr 2021

A quasi-experimental study on stethoscopes contamination with multidrug-resistant bacteria: Its role as a vehicle of transmission

PONE-D-20-26077R1

Dear Dr. Su-Mi Choi,

We’re pleased to inform you that your manuscript has been judged scientifically suitable for publication and will be formally accepted for publication once it meets all outstanding technical requirements.

Kind regards,

Jose Alvarez

Guest Editor

PLOS ONE

---

## [Editor Report · Acceptance letter]

12 Apr 2021

PONE-D-20-26077R1 

A quasi-experimental study on stethoscopes contamination with multidrug-resistant bacteria: Its role as a vehicle of transmission 

Dear Dr. Choi:

I'm pleased to inform you that your manuscript has been deemed suitable for publication in PLOS ONE. Congratulations! Your manuscript is now with our production department. 

Kind regards, 

on behalf of

Dr. Jose Alvarez 

Guest Editor

PLOS ONE